# Navigation Guided Biopsy Is as Effective as Frame-Based Stereotactic Biopsy

**DOI:** 10.3390/jpm13050708

**Published:** 2023-04-23

**Authors:** Dae Hyun Lim, So Yeon Kim, Young Cheol Na, Jin Mo Cho

**Affiliations:** Department of Neurosurgery, International St Mary’s Hospital, Catholic Kwandong University, Incheon 22711, Republic of Korea

**Keywords:** navigation, biopsy, frameless steretaxy

## Abstract

Background: Stereotactic biopsy is a standard procedure for brain biopsy. However, with advances in technology, navigation-guided brain biopsy has become a well-established alternative. Previous studies have shown that frameless stereotactic brain biopsy is as effective and safe as frame-based stereotactic brain biopsy is. In this study, the authors evaluate the diagnostic yield and complication rate of frameless intracranial biopsy. Materials and Methods: We reviewed data from biopsy performed patients between March 2014 and April 2022. We retrospectively reviewed medical records, including imaging studies. Various intracerebral lesions were biopsied. Diagnostic yield and post-operative complications were compared with those of frame-based stereotactic biopsy. Results: Forty-two frameless navigation-guided biopsy were performed, and the most common pathology was primary central nervous system lymphoma (35.7%), followed by glioblastoma (33.3%), and anaplastic astrocytomas (16.7%), respectively. The diagnostic yield was 100%. Post-operative intracerebral hematoma occurred in 2.4% of cases, but it was not symptomatic. Thirty patients underwent frame-based stereotactic biopsy, and the diagnostic yield was 96.7%. There was no difference in diagnostic rates between two methods (Fisher’s exact test, *p* = 0.916). Conclusions: Frameless navigation-guided biopsy is as effective as frame-based stereotactic biopsy is, without causing further complications. We consider that frame-based stereotactic biopsy is no longer needed if frameless navigation-guided biopsy is used. A further study will be needed to generalize our results.

## 1. Introduction

Stereotactic biopsy is a type of minimally invasive medical procedure used to obtain a tissue sample of an abnormality in the brain. Most stereotactic procedures are frame-based, which facilitates precise access and positioning of the biopsy needle within the lesion. Frame-based stereotactic biopsy is a type of stereotactic biopsy that involves the use of a special head frame to precisely target the abnormality being biopsied. This technique is commonly used for brain biopsies, where the abnormality is located deep within the brain and requires a high degree of precision. During a frame-based stereotactic biopsy, the patient’s head is first secured in a rigid frame that is attached to the skull using pins or screws. While frame-based stereotactic biopsy is a highly accurate and effective method for obtaining tissue samples from deep within the brain, there are some potential disadvantages and risks associated with the procedure. 

For instance, one of the disadvantages of frame-based stereotactic biopsy is the pain experienced by the patient during the procedure. The placement of the head frame and pins or screws can be uncomfortable for some patients, and immobilization of the head during the procedure can be stressful or anxiety-inducing. 

Frameless navigation-guided biopsy is an alternative method to frame-based stereotactic biopsy, which uses imaging technology and computer guidance to precisely target and obtain tissue samples from abnormal areas within the body. Frameless navigation-guided biopsy has been proven to be as accurate as frame-based stereotactic biopsy is [1,2,3]. In detail, the diagnostic yield of frameless stereotactic brain biopsy was between 88.9% and 99.7% of cases [4,5,6,7,8,9,10,11,12,13,14]. Furthermore, they can modify the original trajectory to target different areas within the brain during the operation. Earlier navigation systems for frameless biopsy lacked devices for rigid biopsy cannula fixation, which did not allow access to very deep-seated lesions [15]. Technical advances have overcome these obstacles.

We retrospectively reviewed our experiences with a frameless, navigation-guided biopsy system (Striker NAV3i^®^, Stryker Leibinger GmbH & Co. KG Bötzinger Straße, Freiburg, Germany). To our knowledge, there have been several reports about the technical features and results; however, this is the first report indicating that navigation-guided biopsy can replace the stereotactic biopsy system, as technical advances have overcome these obstacles.

## 2. Materials and Methods

This retrospective study was approved by our institutional review board, which waived the requirement for patient informed consent given the retrospective nature of this study (IRB number is IS23RIMI0009). All procedures were in accordance with the ethical standards of the institutional and/or national research committee and the 1964 Declaration of Helsinki and its later amendments or comparable ethical standards.

We retrospectively reviewed a series of 42 patients (24 men and 18 women) with various intracerebral lesions who underwent biopsy. The median age of the patients was 53.9 years (range, 25–81 years). Preoperative planning was performed using magnetic resonance imaging (MRI) data provided by a radiological picture archiving and communication system (PACS), which were sent to the planning station. In the operating room, registration was performed by referencing these morphological data to the position of the patient’s head. After general anesthesia was administered, the Mayfield^®^ head clamp was secured, sterile draping was performed, and image-guided burr hole surgery was performed at the entry point. Since frameless navigation-guided biopsies were performed under general anesthesia, patients did not complain of discomfort. Following burr hole surgery, the frameless guiding system, consisting of three segments (base plate, upper plate, and cannula), was installed on the skull (see Figure 1). After the frameless guiding system was set up, the image-guided biopsy needle was advanced toward the target point. The approach of the needle’s tip toward the target was visualized on the computer screen (see Figure 2). Cautious tissue aspiration was performed, and the inner needle was turned 180 degrees to cut off a cylindrical tissue sample. Post-operative MRI was performed to confirm the success of the biopsy (see Figure 3).

We retrospectively reviewed our series of 30 frame-based stereotactic procedures and compared the results. Most of those who were treated with frame-based stereotactic biopsy were treated before frameless navigation biopsy was performed, and it has been rarely used since its introduction. Data were extracted from our surgical database, and the operation time, biopsy results, and number of adverse events were recorded. We used the Wilcoxon test to compare procedural times between the frame-based and frameless biopsies and Fisher’s exact test to compare the diagnostic rates of both procedures. All statistical analyses were performed using IBM SPSS software version 24.0 (IBM Corp., Armonk, NY, USA), and a *p*-value of 0.05 was considered to be statistically significant.

## 3. Results

The biopsied lesions had a mean maximal diameter of 45.87 mm (range, 1.2–9.5 mm). Forty-two lesions were situated supratentorially and underwent histological examination. During surgery, the samples were sent to the neuropathology department, and the frozen biopsy result was communicated to the surgeon. The procedures were completed after confirmation of the frozen biopsy result. Pathological tissue was obtained in all cases. Histopathology confirmed 15 primary central nervous system lymphomas, 14 glioblastomas, 7 anaplastic astrocytomas, 2 metastatic brain tumors, 1 lymphocytic vasculitis, 1 germinoma, and 1 necrotic tissue of cerebral infarction. In the patient with necrotic tissue of cerebral infarction, another sample was taken from the contrast-enhanced rim, which only revealed brain tissue with a higher than normal amount of necrotic cells, which was diagnosed as cerebral infarction.

Registering the Striker NAV3i^®^ navigation system took from 2 to 3 min. The learning curve was relatively steep. The mean operation time was 45 min (range, 23–68 min), including the time until the first frozen biopsy result was reported. No adverse events occurred during or after the operations.

We biopsied over 30 patients using the Leksell^®^ stereotactic frame. Frame-based stereotactic biopsy was performed under local anesthesia, causing discomfort and additional pain to the patient. The average age was 62 years, and all lesions were supratentorial. Twenty-nine biopsies were diagnostic (96.7%), while one biopsy (3.3%) had no diagnostic yield. There were no adverse events, with one case of post-operative minor bleeding that did not require further treatment. The mean procedural time within the operating room was 47 min. However, preparation with frame adjustment followed by MRI for planning and calculation of the target vectors took another 47 to 55 min, increasing the mean procedural time to 94 min.

Statistically, we did not find a difference comparing the diagnostic rates of both procedures (Fisher’s exact test, *p* = 0.916, Table 1). In both series, no serious adverse events were encountered. Comparison of the procedural times between both techniques (45 min versus 94 min) revealed a statistically significant difference in favor of the frameless procedure (*p* < 0.001, Wilcoxon test).

## 4. Discussion

This study evaluated the effectiveness and safety of frameless navigation-guided brain biopsy compared to those of standard frame-based stereotactic biopsy. The results indicate that frameless navigation-guided biopsy is as effective as frame-based stereotactic biopsy is in terms of diagnostic yield, with a 100% diagnostic yield for frameless biopsy and a 96.7% diagnostic yield for frame-based biopsy. These findings suggest that frameless navigation-guided biopsy may be a suitable alternative to frame-based stereotactic biopsy for intracranial lesions. The authors suggest that the frame-based approach may no longer be necessary if frameless navigation-guided biopsy is used. 

The exact pathological diagnosis is crucial for the treatment of a patient. Therefore, it is necessary to sample a sufficient amount of representative tissue, regardless of the method used for brain biopsy. Until now, the gold standard for stereotactic brain biopsy has been frame-based stereotactic brain biopsy [5]. However, there are some potential disadvantages and risks associated with frame-based stereotactic brain biopsy. The placement of the head frame and pins or screws can be uncomfortable for some patients, and immobilization of the head during the procedure can be stressful or anxiety-inducing.

Frameless navigation-guided biopsy is an emerging neurosurgical procedure that has shown relatively better outcomes compared to those of frame-based techniques [7]. Unlike frame-based stereotactic biopsy, which requires the use of a rigid head frame, frameless navigation-guided biopsy does not require the patient’s head to be immobilized. Such as frame-based stereotactic biopsy, navigation-guided biopsy is minimally invasive and typically has a lower risk of complications and a shorter recovery time than traditional open surgery does. This can make the procedure more comfortable for the patient. Such as frame-based stereotactic biopsy, frameless navigation-guided biopsy is minimally invasive and typically has a lower risk of complications and a shorter recovery time than traditional open surgery does. 

In this study of 42 cases of frameless navigation-guided biopsy using the frameless guiding system and 30 patients who underwent frame-based stereotactic biopsy, we found a pathological yield rate of 100%, which is not statistically different, but it is considered to be a slightly superior to that of frame-based stereotactic biopsy (96.7%). Navigation technology allows the precise targeting of abnormal tissue, potentially reducing the risk of false-negative or false-positive results. According to the literature, the diagnostic rates for both frame-based and frameless procedures ranged from 89% to 99%, and there were no significant differences between the techniques [4,7,10,14,16,17,18,19].

Accurately positioning the tool tip in a frameless stereotactic biopsy has always been an important issue. Rigid biopsy cannula fixation devices needed for this purpose are also used in other biopsy systems, such as the Vertek^®^ system (Medtronic, Minneapolis, MN, USA). A laboratory study with another device (VarioGuide^®^, Feldkirchen, Germany) demonstrated a mean deviation from the target point of 0.7 mm and 100% target localization in spherical lesions of 0.524 cm^3^, confirming the high accuracy of the system [17]. 

Interestingly, the procedural time for using our frameless guiding system was significantly shorter in comparison with that of our frame-based technique, and there were no complaints of pain during the frame fixation procedure.

We prefer the frameless guiding system for biopsies for several reasons. Firstly, the frameless guiding system is performed under general anesthesia, which means there is no pain and it is well tolerated. Although the frame-based procedure is performed under local anesthesia, the patient may experience pain, which can occasionally make the procedure impossible to perform.

Secondly, the frameless guiding system allows the referencing of the navigation system using surface data from the face and does not necessarily require a new imaging with fiducials.

Lastly, compared to frame-based procedures, the frameless guiding system has the advantages of 3-dimensional visualization of the approach of the biopsy needle towards the lesion and the possibility to take biopsies from targets other than the preplanned targets within the same operative procedure without new plans.

Regarding accuracy, both methods are comparable [20,21,22]. However, the overall clinical application accuracy is lower in the frameless system due to the lack of satisfactory mechanical cannula guidance. A review of both techniques has been published [22]. Factors such as image acquisition modality and brain shift can affect both frameless and frame-based methodologies.

However, our study has several limitations. 

First, this study design is retrospective, which means that data were not collected in a systematic and standardized manner. This could potentially introduce bias into the results, as well as limit the ability to make causal inferences.

Second, this study includes a relatively small sample size of 42 navigation-guided brain biopsy procedures. A larger sample size may be needed to confirm the findings and increase the generalizability of the study.

Last, the patients who underwent navigation-guided brain biopsy may have been selected based on time sequence, which may not apply to all patients who require brain biopsy. This may limit the generalizability of the findings.

Overall, the results suggest that frameless navigation-guided biopsy is a safe and effective alternative to frame-based stereotactic biopsy for intracranial lesions. The reduced invasiveness and shorter procedure time associated with frameless biopsy may offer significant advantages over frame-based biopsy, and the high diagnostic yield observed in the study indicates that this approach may be a suitable option for a wide range of brain lesions. However, as with any medical procedure, further research is needed to confirm the generalizability of these findings and to identify any potential limitations or complications associated with frameless navigation-guided biopsy.

## 5. Conclusions

Our results confirm that frameless, navigation-guided biopsy is a safe and successful method for obtaining a pathological diagnosis even in deep-seated lesions. In our opinion, frameless navigation-guided brain biopsy offers many advantages over frame-based stereotactic biopsy. Therefore, we believe that frame-based stereotactic biopsy is unnecessary. Future studies will be needed to generalize our results.

## Figures and Tables

**Figure 1 jpm-13-00708-f001:**
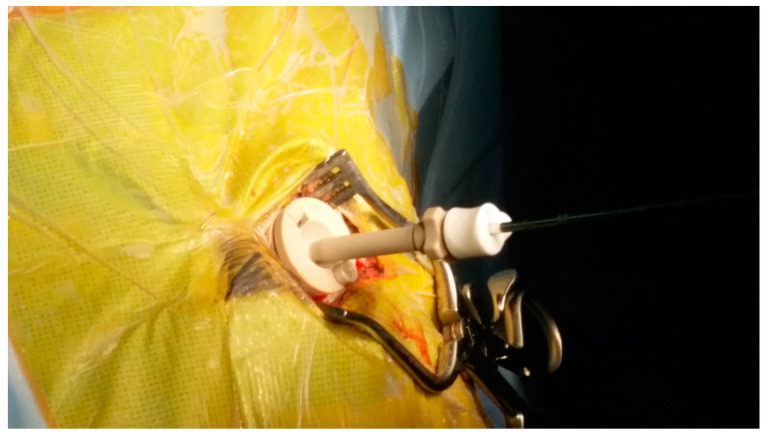
Frameless guiding system was mounted on the skull.

**Figure 2 jpm-13-00708-f002:**
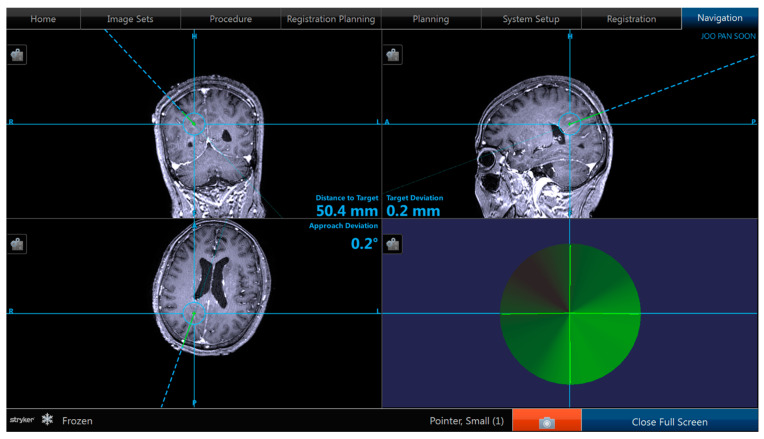
Snap shot of the navigation image.

**Figure 3 jpm-13-00708-f003:**
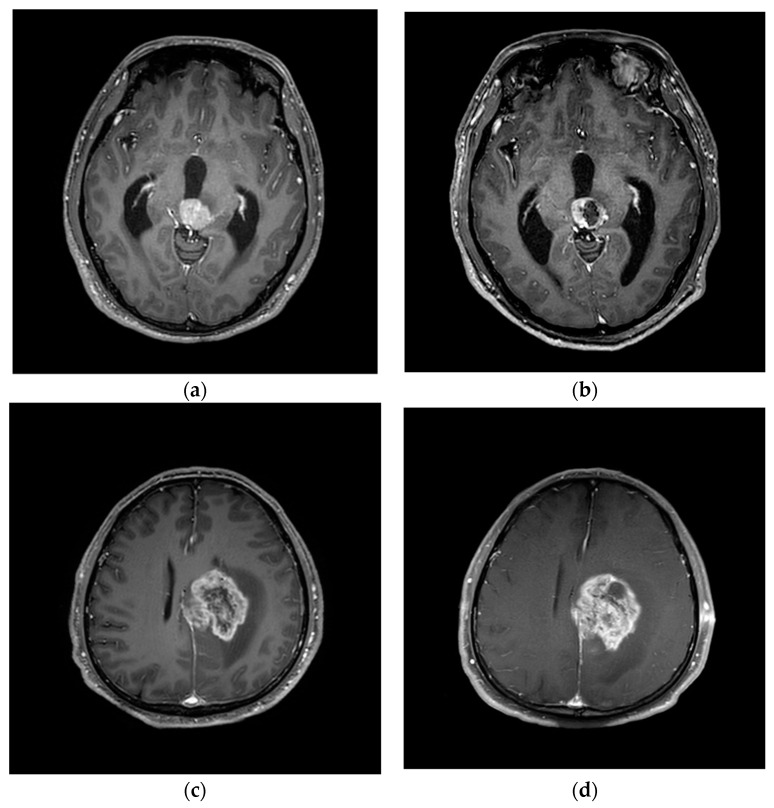
Pre-operative image (**a**,**c**) and post-operative image (**b**,**d**). Post-operative image confirmed the location of biopsy.

**Table 1 jpm-13-00708-t001:** Comparison of clinical, neuroimaging, and operative variables between frame-based and frameless stereotactic biopsy procedures.

No. (%)			
Variable	Framelss	Frame-Based	*p* Value
no of procedure	42	30	
male sex	19 (45.2)	17 (56.7)	0.896
mean Max lesion diameter (mm)	45.87 mm (1.2–9.5 mm)	36.85 (1.9–8.6 mm)	0.865
diagnostic yield	42 (100)	29 (96.7)	0.916

## Data Availability

Not applicable.

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
