# Peer review of "Navigation Guided Biopsy Is as Effective as Frame-Based Stereotactic Biopsy"

_jpm, 2023, doi:10.3390/jpm13050708_

Round 1

Reviewer 1 Report

The Authors present their experience with frameless guided biopsy, reporting on 42 patients treated in 8 years, and compared to a sample of frame-based biopsies of 30 patients.

The paper is of some interest, but I have a few suggestions before publication might be considered:

1. it is not clear when the frame-based patients were treated; 

2. has the work been approved by the local ERB?

3. the Authors state that the advantage of frameless navigation is not using pins, but they report using the Mayfield clamp.

The paper should be more concise, especially in the Discussion section, where multiple concepts are elaborated more than once.

Author Response

The Authors present their experience with frameless guided biopsy, reporting on 42 patients treated in 8 years, and compared to a sample of frame-based biopsies of 30 patients.

The paper is of some interest, but I have a few suggestions before publication might be considered:

  1. it is not clear when the frame-based patients were treated; 

Most of those treated with frame-based therapy were treated before those treated with navigation guided therapy. This part has been described.

  1. has the work been approved by the local ERB?

Yes we have received an irb from our institution and the number is IS23RIMI0009. This part has also been described.

  1. the Authors state that the advantage of frameless navigation is not using pins, but they report using the Mayfield clamp

Mayfield pins are driven under general anesthesia, so there is no additional pain for the patient. However, frame-based patients experience additional discomfort as the pins are driven under local anesthesia. This part has also been described. 

.

The paper should be more concise, especially in the Discussion section, where multiple concepts are elaborated more than once.

I've tried to be more concise. ^^;

Reviewer 2 Report

Brain biopsy is a surgical procedure in which a small amount of intracranial tissue is obtained for pathological analysis. The manuscript by Dae Hyun Lim et al compared two methods of obtaining biopsy: frame-based biopsy and frameless stereotactic brain biopsy, demonstrating similar results for diagnostic yield for both systems. However, the manuscript needs several improvements before it can be considered for publication.

  1. The authors did not include the consensus from the competent Institutional Ethics Committee in the manuscript.
  2. The legends lack information, and the authors should include more details.
  3. The results obtained should be shown in tables with relative statistical analysis (p-value).
  4. It would be interesting to compare the two methods, taking into consideration the size and depth of the lesions.
  5. The table should be removed from the main text and included in another section of the manuscript.
  6. In the introduction there are words with different font size.
  7. In the discussion, the authors should highlight the novelty of their results.

Author Response

Brain biopsy is a surgical procedure in which a small amount of intracranial tissue is obtained for pathological analysis. The manuscript by Dae Hyun Lim et al compared two methods of obtaining biopsy: frame-based biopsy and frameless stereotactic brain biopsy, demonstrating similar results for diagnostic yield for both systems. However, the manuscript needs several improvements before it can be considered for publication.

  1. The authors did not include the consensus from the competent Institutional Ethics Committee in the manuscript.

we have received an irb from our institution and the number is IS23RIMI0009. This part has also been described.

  1. The legends lack information, and the authors should include more details.

What do you mean by legend? I think I described it in detail...

  1. The results obtained should be shown in tables with relative statistical analysis (p-value).

we have marked

  1. It would be interesting to compare the two methods, taking into consideration the size and depth of the lesions.

The size and depth of the lesions varied and were not compared ^^*

  1. The table should be removed from the main text and included in another section of the manuscript.

I think the table should be there because it summarizes the clinical data. If we need to remove it, we will remove it.

  1. In the introduction there are words with different font size.

The font size has been unified.

  1. In the discussion, the authors should highlight the novelty of their results.

The novelty of the research findings was emphasized. please look again

Round 2

Reviewer 1 Report

the paper has been revised only partially. 

Author Response

I agree with the reviewer's opinion. Removed the things to be removed and added the things to be added. If you need more, please let me know and I will fix it.

Reviewer 2 Report

The authors need to carefully consider and address all reviewer comments. At this stage the manuscript is not suitable for publication.

Author Response

(The authors gave the same response as above.)
